# Collaborative optimization of multi-modal transport solutions for urban-rural bus routes

Jun Zhang [1]*, Jingyi Qin[1], Jinliang Shao[1], Jiajun Shen[1], Bin Lv[2]

1 Department of Transportation Engineering, College of Architectural Science and Engineering, Yangzhou University, Yangzhou, Jiangsu Province, China, 2 Comprehensive Administrative Law Enforcement Detachment, Transportation Bureau of Yangzhou, Yangzhou, Jiangsu Province, China

* Zhangjun93@yzu.edu.cn

**Data Availability Statement:** All relevant data are within the manuscript and its Supporting Information files.

**Funding:** Funder statement: The funders had no role in study design, data collection and analysis,

## Abstract

Faced with the land use heterogeneity and trip distribution non-equilibrium along the corridor connecting urban district and rural area, the operation characteristics of the all-stop bus, the express bus and the shuttle bus running on the transportation corridor should be given full consideration, as well as the tradeoff between passenger trip cost and bus operation cost during the peak-hour period. In order to realize the reliable decision of bus resources configuration for urban-rural public transportation, a bi-objective programming model of multi-modal urban-rural bus operation scheme is established, where the decision variables include the departure frequency of different bus modals, the terminals of shuttle buses, and the stop scheme of express buses, under the constraints of departure interval, passenger volume, vehicle occupancy, section capacity usage and modal competition. To validate the feasibility of proposed optimization model, an urban-rural bus corridor from Rural Jasmine Ecological Park in Hanjiang District to West Xiangyang Bridge in Songqiao Town was chosen as a study case, and the operation organization scheme of three bus modals were solved by the mathematical solver Lingo using mixed integer programming. Meanwhile, the scheme differences under different subobjective weights were discussed, taking into account the uncertain passenger demand and operation supply. The result shows that the increase of passenger trip cost weight will increase the departure frequency of all three bus modals, while the increase of operation cost weight will decrease the total operating vehicles, but the stop frequency at middle stations for express buses will increase accordingly. Compared to the current operation scheme, the optimized scheme can greatly enhance the section capacity usage at a slight sacrifice of passenger trip time with fewer bus vehicles.

## Introduction

In China, the increasing urbanization process has accelerated the population mobility between urban areas and rural areas, as evidenced by the growing demand of urban-rural trips. Due to the increasing urban-rural transportation connection, the development planning of comprehensive transportation issued by the State Council of China has explicitly requested to promote rural vitalization by strengthening the urban-rural transportation connection, in order to

decision to publish, or preparation of the manuscript. Funding statement: 1, Project of National Natural Science Foundation of China under Grant number 52302395. Recipient: Jun Zhang; 2, MOE (Ministry of Education in China) Project of Humanities and Social Sciences under Grant number 23YJAZH122. Recipient: Jiajun Shen.

**Competing interests:** The authors have declared that no competing interests exist.

facilitate the resource sharing of passenger transportation and the construction of modern urbanization. Compared to urban buses, urban-rural buses possess the following characteristics:

(1) The average trip distance is longer. Due to the differences in land use and population density, there are only a few passenger distributing stations along the urban-rural corridors, the average trip distance of urban-rural bus is longer than the urban bus. Meanwhile, most stations on the urban-rural corridors have fallen into disuse, leading to the increase of actual stop spacing.

(2) Passenger trips are more concentrated during the peak hour period. Since the urban-rural bus mainly serves the commuting trips along the urban-rural corridor, the passenger volume during rush hours usually account for 50% of daily trips. While the rush hour volume on urban bus usually accounts for 15 to 20% of daily trips, because there still exist large passenger volumes with the purposes of business travel, medical travel, recreation travel, shopping travel, tour travel during the non-peak hours.

(3) The directional volume imbalance is more obvious. For the urban bus corridor, there is no significant difference between the two directional volumes during the same time period. While the tide traffic along the urban-rural corridor leads to an apparent direction volume disequilibrium, e.g., in the morning peak hour of workdays, the number of passengers towards the urban area is bigger than the passengers from urban district to urban area.

At the moment, the urban-rural bus transit is stuck with monotonous service modal and lower passenger coverage, and traditional all-stop buses and request-stop buses are suffering from the problem of a lower punctuality and a higher vehicle running empty [1]. Meanwhile, Residents with long-distance trips and those waiting at high-volume stations have a longer on-board time and waiting time under the single all-stop modal. For the sake of enhancing the service quality and passenger attraction of urban-rural bus transit, it is of urgent need to study the optimization method for urban-rural bus operation. Current literatures on bus transit management can be divided into the optimization of operation parameters and the optimization of bus service modals.

In terms of bus operation parameters optimization, extensive studies [2, 3] have been performed on the parameters including departure frequency, vehicle configuration, vehicle type selection, vehicle routes, stop scheme and etc. Eriksson et al. found that marginally adjusting the departure time can reduce the vehicle use, and the saved cost could be helpful in subsidizing long-run bus lines [4]. However, under the dynamic fluctuation of passenger volume, the passengers' perception of crowdedness should be considered during scheme optimization. Therefore, Pei et al. further studied the impact of on-board crowdedness on the boarding decision of station waiting passengers to quantify the boarding and alighting number of station passengers, and established an elastic departure interval decision model aiming at the minimization of travelling cost, waiting cost and operating cost [5]. Uunder the limitations from transportation resources and historical reasons, Sadrani et al. formulated a waiting time optimization model for the mixed-fleet dispatching of buses, where 12-m long, 15-m long and 18-m long bus types were considered, and results show that the uneven headways outperform the even headway on a heavy-demand bus corridor [6]. Recently, for the sake of suiting the demand and behavior changes in the post-covid-19 pandemic, Filgueiras et al. optimized the bus network frequency in coordination with the transit network reconstruction [7].

With the development of demand responsive buses, Dytckov et al. built a simulator composed of traveler model, service model and emission model, where mode choice model and vehicle routing algorithm are embedded [8]. Targeting at the maximization of operation profit and social welfare, Ma et al. proposed a pricing model for the flexible bus service, with special considerations on the passenger perceived time utility and acceptance of detours [9];

Takamatsu and Taguchi put forward a programming model to output a bus timetable with less transfer waiting time in the rural area [10]. In order to save the system power supply, Nitta et al. designed an intelligent bus stop system by applying the normally-off strategy upon revealing the frequency of use and the passenger volume [11].

As to the optimization of bus service modals, the studies concerning bus modals usually include three types, the all-stop bus modal, the shuttle bus modal (also known as the short-turning modal) and the express bus modal (also known as the skip-stop modal) [12]. During the study of all-stop service modal, Herbon et al. gave extra considerations to the passenger crowding cost and empty seat cost when minimizing the cost of passengers and operators, in order to solve the lower capacity usage under the fixed departure interval [13]; Zhang et al. took into account the factors of vehicle economy, peak-hour capacity usage and fleet scale in the optimization model, where the departure time and vehicle type of each bus [14]. Based on the level of service and operation cost, Pena et al. proposed an optimization model of hybrid vehicle type configuration [15]. As regards the service modals of shuttle bus and express bus, they are frequently combined with the studies of all-stop bus modal to realize a collaborative optimization. For instances, Gkiotsalitis et al. validated the feasibility of running additional shuttle buses besides the all-stop buses, which can better satisfy the demand spatial distribution during the peak period [16]; Cao and Ceder introduced the skip-stop tactic to shuttle bus dispatching and devised an integrated optimization of service timetabling and bus scheduling, aimed at reducing both travel time and vehicle usage [17]. By incorporating the multiple timetabled services into the bus scheduling along the intercity bus corridor, Steiner formulated a programming model by setting up decision variables such as departure time, stop plan, passengers served and service duration [18]. Similarly, Ji et al. designed a coordinated optimization technology of multiple bus services, via a bi-level programming from perspectives of passenger trip time and operator cost [19]. Moreover, in order to save the energy cost, Tang et al. proposed an improved scheduling model for single bus line by employing the skip-stop strategy [20].

From the foregoing literatures, it is observable that current researches into the optimization objectives, model constraints and solution algorithms for bus transportation organizations have established a well-developed system of methodologies and technologies, while the major application scenarios focus on the urban or suburban area, and the emerging services of customized shuttle buses and flexible buses are confined to the commuting routes and hub connection lines, where the flexible buses require higher accuracy in dynamic passenger volume prediction [21]. Faced with the disequilibrium of passenger trip distribution along the urban-rural transit corridor, the current research gap is in integrating and coordinating the operation scheme of various bus service modals. Aiming at the compromise between passenger travel convenience and operator cost control, the major contribution of this paper lies in the formulation of a mixed integer programming model for multi-modal bus operation optimization, with special consideration of demand adaptability and economic feasibility, in order to synchronously optimize the service frequency, vehicle configuration and stop plan of different bus modals.

The remainder of this paper is structured as follows. By introducing the service features of three representative bus service modals, section *Collaborative modelling* formulates the multi-modal bus optimization problem as an MIP model upon reasonable assumptions. The *Solution procedure* section illustrates the solving method and algorithm of proposed joint optimization model, and the *Model validation and discussion* section presents a comprehensive analysis from parameter calibration to result discussion, using an urban-rural bus corridor in Yangzhou, Jiangsu Province, China as an illustration. Finally, the *Conclusion* section concludes the paper by discussing the main highlights and possible future work.

## Collaborative modelling

### Problem description and assumptions

**Problem description.** The purpose of this paper is to solve the mixed operation of different bus modals running on the urban-rural corridor, in order to provide feasible decision schemes of modal choice, departure frequency, stop plan, running section and etc., considering the following features of three typical bus service modals. Table 1 indicates the current operation schemes of three bus modals.

(1) The all-stop bus is a common modal in bus transit operation, where the bus vehicles run between two terminals and dwell at every stop, with the advantage of offering equal services for passengers at every station. However, the comfortability of passengers travelling long trip distances will be reduced by the increased number of stops. A bus route with an equitable distribution of station volumes is accommodated by the all-stop bus modal.

(2) The express bus is a kind of stop-restricted operation modal, where the bus vehicles only dwell at stations with larger passenger alighting and boarding demand, with the advantage of reducing stop frequency and passengers trip time. The express bus modal is adopted in the bus route with higher differences in station passenger demands.

(3) The shuttle bus runs along a proportion of the full line between turn-back stations, where the two turn-back stations can be intermediate stations or one intermediate station and one terminal station. The shuttle bus modal helps in balancing the section volume distribution and enhancing the vehicle capacity usage.

In the subsequent modelling, only one operation direction is considered, since the bus operation on each running direction can be regarded as an independent optimization problem considering the following factors.

(1) During the same time period, there appears a great difference in the passenger volumes on two directions. E.g., when the hourly passenger volume of one direction reaches its peak point, the volume on the other direction is usually less than one-third of the peak volume. Therefore, the bus operation schemes of two directions should be optimized separately under different demand inputs, otherwise the bus vehicle capacity on the low-volume direction will be excessively wasted, or the basic level of service on the high-volume direction couldn't be guaranteed.

(2) The upstream and downstream passenger volumes show a strong distribution heterogeneity with each other. On one hand, the stations along the two directions are not identical to each other, namely the number of stations, the station location, and the distribution of station spacing are different. On the other hand, the high-volume sections on the two directions are different.

**Table 1. Characteristics of representative bus routes along the urban-rural transportation corridor.**

| Service model | Bus route | Operation period | Average departure interval (min) | Average stop spacing (m) |
|---|---|---|---|---|
| Suttle bus | 15 | 6:10–18:35 | 14 | 549 |
| | 15 (night) | 16:33, 17:23, 18:11 | 49 | 549 |
| | 105 (night) | 18:45–21:40 | 25 | 584 |
| Rapid bus | Youyang sightseeing | 7:20–18:30 | 30 | 2692 |
| | Songyang express | 5:40–17:30 | 30 | 3701 |
| All-stop bus | 103 | 5:40–17:55 | 19 | 499 |
| | Guoyang trunk | 6:00–18:40 | 60 | 919 |

### Basic model assumptions

The following reasonable assumptions are supposed to better establish the mathematic programming model.

Assumption 1: The arrival rate of station passengers obeys a uniform distribution, whereby the average passenger waiting time at a station is the half of stopped bus arriving interval [22].

Assumption 2: The bus vehicles under all-stop, express and shuttle modals depart at different frequencies during the peak hour.

Assumption 3: Passengers choose the appropriate bus service modal that meets their destination station stop plan, disregarding the transfer behavior between different service modals.

Assumption 4: The vehicles of different bus modals are all powered by electricity, and the running cost is measured by unit electricity consumption.

Assumption 5: Since the station volume along the urban-rural bus route is relatively small, the alighting time and boarding time are counted in the fixed bus dwelling time.

Assumption 6: As a public-benefit transit, the ticket price of urban-rural bus service is usually fixed, and therefore the item of ticket cost is ignored under the given passenger demand.

### Modelling framework

The framework of formulating the problem is illustrated in Fig 1. Considering the uncertainty of service modal pricing and the complexity of bus cost, the passenger-oriented objectives are based on travel time, and the operator-oriented objectives are measured by the operation cost. The stop plan of express bus, modal departure frequency and the shuttle bus terminals are the decision variables.

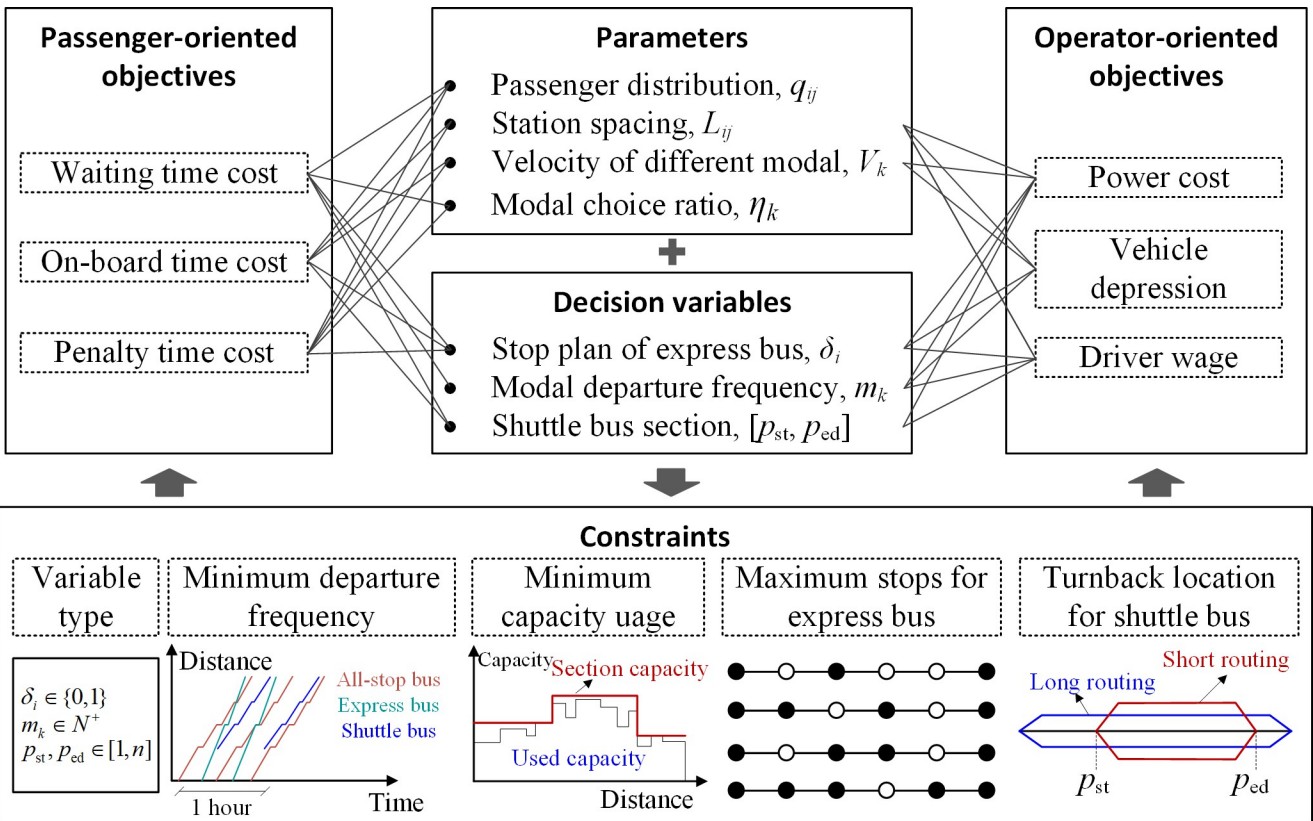

**Fig 1. Model formulation framework.**

## Objective functions

The formulation of optimal objectives considers both passenger travel time cost and operator service cost. The travel time cost includes the waiting time cost, the on-board time cost the penalty cost, which will be explained later. The operator service cost considers the vehicle energy consumption, the human driving cost and etc.

## Passenger travel time cost

In addition to the normal station waiting time cost and passenger on-board time cost, the penalty cost has been given a special consideration under the combined operation of three bus modals. The penalty cost refers to the additional trip time cost arising from the station passengers, who have to choose the all-stop bus because their station is skipped by the express bus. The objective of passenger trip time cost is formulated as

$$C_1 = C_{wait} + C_{in} + C_{pe} = vot \times (T_{wait} + T_{in} + cop \times T_{pe}) \tag{1}$$

Where $C_{wait}$, $C_{in}$ and $C_{pe}$ represent the passenger waiting time cost, the on-board time cost and the penalty cost respectively, Yuan; $vot$ denotes the average value of time of bus passengers, Yuan/(pax·min); $T_{wait}$, $T_{in}$ and $T_{pe}$ refer to the passenger waiting time cost, the on-board time and the potential delayed time respectively, min; $cop$ denotes the penalty coefficient.

*(1) Passenger waiting time.* Under assumption 2, the departure interval is fixed for a single bus service modal during a specific period. Considering the spatial distribution along the shuttle bus running sections, the calculation of waiting time can be divided into two parts.

1) The passenger waiting time at the stations passed by all-stop and express bus vehicles ($T_{wait}^{QD}$) can be measured by

$$T_{wait}^{QD} = \sum_{i=1}^{p_{st}-1}\left[\sum_{j=i+1}^{n}q_{ij}\left((1-\delta_i\eta_2)\frac{30}{m_1}+\delta_i\eta_2\frac{30}{m_2}\right)\right]+$$
$$\sum_{i=p_{ed}}^{n-1}\left[\sum_{j=i+1}^{n}q_{ij}\left((1-\delta_i\eta_2)\frac{30}{m_1}+\delta_i\eta_2\frac{30}{m_2}\right)\right] \tag{2}$$

Where $p_{st}$ and $p_{ed}$ represent the index number of the starting station and the ending station of the shuttle bus; $n$ is the total number of stations on current bus transit line; $q_{ij}$ denotes the trip volume from station $i$ to station $j$, pax/h; $\delta_i$ is the 0–1 binary decision variable of stop plan at station $i$ under the express bus modal, where 1 denotes a stop and 0 denotes a skip; $\eta_2$ represents the ratio of passengers choosing to take express bus vehicles; $m_1$ and $m_2$ denote the departure frequency of all-stop bus vehicles and express bus vehicles respectively, veh/h.

2) The passenger waiting time at the stations passed by three bus modal vehicles ($T_{wait}^{QDJ}$) can be measured by

$$T_{wait}^{QDJ} = \sum_{i=p_{st}}^{p_{ed}-1}\left[\sum_{j=i+1}^{p_{ed}}q_{ij}\left((1-\delta_i\eta_2-\eta_3)\frac{30}{m_1}+\delta_i\eta_2\frac{30}{m_2}+\eta_3\frac{30}{m_3}\right)\right] \tag{3}$$

Where $\eta_3$ is the ratio of passengers choosing to take shuttle bus vehicles; $m_3$ denotes the departure frequency of shuttle bus vehicles, veh/h.

Combining Eq (2) and Eq (3), the total passenger waiting time at all bus stations can be obtained by

$$T_{wait} = T_{wait}^{QD} + T_{wait}^{QDJ} \tag{4}$$

*(2) Passenger on-board time.* The on-board time of bus passengers consists of the section running time, the dwelling time (including the stop time and the speed loss time caused by acceleration and deceleration). The passenger on-board time is also calculated on the basis of divided section types, as indicated in Eq (5) and (6).

1) The on-board time during the sections covered by all-stop and express bus vehicles ($T_{in}^{QD}$) can be measured by

$$
\begin{aligned}
T_{in}^{QD} &= \sum_{i=1}^{p_{st}-1}\sum_{j=i+1}^{n} q_{ij}t_{ij} + \sum_{i=p_{st}}^{p_{ed}-1}\sum_{j=i+1}^{n} q_{ij}t_{ij}, \\
t_{ij} &= (1 - \delta_i\eta_2)\left(\frac{60L_{ij}}{V_1} + \sum_{s=i+1}^{j}\frac{t_s}{60}\right) + \delta_i\eta_2\left(\frac{60L_{ij}}{V_2} + \sum_{s=i+1}^{j}\delta_s\frac{t_s}{60}\right)
\end{aligned}
\tag{5}
$$

Where $t_{ij}$ denotes the sum of section running time and station dwelling time from station $i$ to $j$, excluding the dwelling time at station $i$, min; $L_{ij}$ is the running distance between station $i$ and station $j$, km; $t_s$ is the additional time consumption caused by stopping at station $s$, including the speed loss time and station dwelling time, s; $V_1$ and $V_2$ represent the average section running speed of the all-stop bus and the express bus respectively, km/h.

2) The on-board time during the sections covered by three bus modal vehicles ($T_{in}^{QDJ}$) can be measured by

$$
T_{wait}^{QDJ} = \sum_{i=p_{st}}^{p_{ed}-1}\sum_{j=i+1}^{p_{ed}} q_{ij}\left[ \begin{array}{l} (1 - \delta_i\eta_2 - \eta_3)\left(\dfrac{60L_{ij}}{V_1} + \displaystyle\sum_{s=i+1}^{j}\dfrac{t_s}{60}\right) + \\[4mm] \delta_i\eta_2\left(\dfrac{60L_{ij}}{V_2} + \displaystyle\sum_{s=i+1}^{j}\delta_s\dfrac{t_s}{60}\right) + \eta_3\left(\dfrac{60L_{ij}}{V_3} + \displaystyle\sum_{s=i+1}^{j}\dfrac{t_s}{60}\right) \end{array} \right]
\tag{6}
$$

Where $V_3$ denotes the average running speed of shuttle buses, km/h.

*(3) Passenger potential delay time.* Passengers who require express bus service at the station that is not served by express buses have to take the all-stop bus, which will result in an additional trip time, namely the potential delay. The measurement of potential delay is

$$
T_{pe} = \sum_{i=1}^{n-1}\sum_{j=i+1}^{n} q_{ij}\left[(1 - \delta_j)\eta_2\left(\frac{60L_{ij}}{V_1} - \frac{60L_{ij}}{V_2}\right)\right]
\tag{7}
$$

**Vehicle operating time cost.** The operator service cost is defined as the product of vehicle running time and unit time operation cost, where the unit time operation cost is calculated considering electricity consumption, drivers wage and vehicle depreciation, as indicated in Eq (8).

$$
C_2 = \sum_{k=1}^{3} coe_k m_k T_k^J, \quad coe_k = \varphi \cdot con_k + de_k + dr_k
\tag{8}
$$

Where $coe_k$ is the unit time operation cost of the $k^{\text{th}}$ bus service modal, $k = 1,2,3$ in turn corresponds to the modal of all-stop bus, express bus and shuttle bus; $m_k$ denotes the departure

frequency of modal $k$, veh/h; $T_k^J$ denotes the running time of modal $k$, h; $\varphi$ is the standard electrovalence, Yuan/KWh; $con_k$ refers to the average electricity consumption of vehicles of modal $k$; $de_k$ and $dr_k$ are the unit time vehicle depreciation and the hourly driver wage under modal $k$, Yuan/(veh·h).

Given the length of the route, the average section running speed and the stop plan of three bus modals, the operator service cost can be formulated as

$$C_2 = coe_1 m_1 \left( \frac{L}{V_1} + \frac{(n-1)t_s}{3600} \right) + coe_2 m_2 \left( \frac{L}{V_2} + \sum_{k=2}^{n} \frac{\delta_k t_k}{3600} \right) + $$
$$coe_3 m_3 \left( \frac{L_{P_{st}}^{P_{ed}}}{V_3} + \frac{(p_{ed} - p_{st})t_s}{3600} \right) \tag{9}$$

Where $L_{P_{st}}^{P_{ed}}$ denotes the one-way running distance of shuttle bus vehicles between its turn-back stations $p_{ed}$ and $p_{st}$.

## Model constraints

In order to guarantee the feasibility and economy of output schemes, special consideration has been given to the stop frequency and the location of shuttle bus turn-back stations, apart from the departure frequency, demand satisfaction and section capacity usage.

**(1) Departure frequency constraint.** To ensure the maximum section volume demand is met, it is important to have a departure frequency that provides the basic bus transportation capacity. As indicated in the Minimum departure frequency module in Fig 1, the capacity varies in spatial sections. For the multi-modal bus transit corridor, the section with the maximum passenger volume is located in the running segments of shuttle buses, hence we have the constraint

$$\sum_{k=1}^{3} b_k m_k \geq \max \left\{ P_{i,i+1} \middle| P_{i,i+1} = \sum_{s=1}^{i} \sum_{j=i+1}^{n} q_{sj}, i \in [p_{st}, p_{ed}] \right\} \tag{10}$$

Where $b_k$ denotes the specified vehicle passenger capacity of the $k^{\text{th}}$ bus service modal, pax/veh; $P_{i,i+1}$ denotes the section passenger volume from station $i$ to station $i+1$ during the peak hour, pax/h.

For the sections that are exclusive to the all-stop bus and the express bus, the departure frequency of the two service modals should satisfy

$$b_1 m_1 + b_2 \delta_s m_2 \geq P_{s,s+1}, s \in [1, p_{st} - 1] \cup [p_{ed} + 1, n] \tag{11}$$

Besides, the departure frequency of each service modal should be a positive integer value, constrained by Eq (12).

$$m_k \in N^+, \ k = 1, 2, 3 \tag{12}$$

**(2) Section capacity usage constraint.** From the perspective of operator, a higher vehicle departure frequency means a higher operation cost. Therefore, it is necessary to improve the section capacity usage when the passenger demands are met. However, passengers' waiting time is being sacrificed due to the increased use of section capacity. According to the study of Yu et al., the vehicle capacity usage ranges from 55.4% to 69.2% [23]. Considering the above-mentioned features and the actual operation data, the lowest section capacity usage is set as

0.6 in our model, as shown in Eq (13).

$$\begin{cases} \dfrac{P_{s,s+1}}{m_1 b_1 + \delta_s m_2 b_2} \geq 0.6, s \in [1, p_{st} - 1] \cup [p_{ed} + 1, n] \\ \dfrac{P_{s,s+1}}{m_1 b_1 + \delta_s m_2 b_2 + b_3 m_3} \geq 0.6, s \in [p_{st}, p_{ed}] \end{cases} \tag{13}$$

**(3) Stop frequency and turnback location constraints.** The stop plan of express bus modal and the turnback stations of shuttle bus modal are key decision variables in the proposed model. Since the stop plans of all-stop bus and the shuttle bus are fixed, the stop frequency constraint is formulated for the express bus to guarantee its rapidness. Eq (14) limits the maximum stop frequency of express buses by taking into account their cumulative station stops, excluding the stops at both terminals.

$$\begin{cases} \delta_s \in [0, 1], \forall s \in [2, n-1] \\ \sum_{s=2}^{n-1} \delta_s \leq \lfloor \dfrac{L}{d_a} \rfloor \end{cases} \tag{14}$$

The turnback stations of shuttle bus modal should be constrained by section passenger volume distribution, where the aggregate passenger volumes in section ($P_{st}$, $P_{st}+1$) and section ($P_{ed}$-1, $P_{ed}$) are surpassed by an empirical value that combines the average passenger volumes of each section and twice the capacity of the shuttle bus vehicle ($b_3$), as indicated in Eq (15).

$$\begin{cases} \sum_{i=1}^{p_{st}} \sum_{j=p_{st}+1}^{n} q_{ij} \geq \dfrac{1}{n-1} \sum_{i=1}^{n-1} P_{i,i+1} + 2b_3 \\ \sum_{i=1}^{p_{ed}-1} \sum_{j=p_{ed}}^{n} q_{ij} \geq \dfrac{1}{n-1} \sum_{i=1}^{n-1} P_{i,i+1} + 2b_3 \end{cases} \tag{15}$$

**(4) Volume conservation constraint under competition.** For the stations passed by all-stop and express bus vehicles, the stop plan of the express bus at stations should satisfy constraint (16). The first item on the left side represents the number of passengers at station $i$ boarding the express bus, only when the express bus stops at station $j$ (namely when $\delta_j = 1$) and the trip volume from station $i$ to station $j$ exists, station $i$ will generate the passengers choosing express buses. The second item on the left side denotes the number of passengers boarding the all-stop bus, and the item on the right side is the aggregate boarding passengers at station $i$.

$$\sum_{j=i+1}^{n} \delta_j q_{ij} \eta_2 + \sum_{j=i+1}^{n} q_{ij}(1 - \eta_2) = \sum_{j=i+1}^{n} q_{ij}, \ i \in [1, p_{st} - 1] \cup [p_{ed} + 1, n] \tag{16}$$

Similarly, for the stations passed by three bus modal vehicles, the stop plan for the express bus at station $i$ should satisfy:

$$\sum_{j=i+1}^{n} \delta_j q_{ij} \eta_2 + \sum_{j=i+1}^{n} q_{ij} \eta_3 + \sum_{j=i+1}^{n} q_{ij}(1 - \eta_2 - \eta_3) = \sum_{j=i+1}^{n} q_{ij}, \ i \in [p_{st}, p_{ed}] \tag{17}$$

## Solution procedure

### Weighted objective establishment

The Max-Min normalization method is employed to eliminate the dimensions differences between passenger trip time cost and operator service cost before linear weighting. As

indicated in Eq (18), the minimum value of each objective is solved by independent optimization, and the maximum values are calculated as twice the minimum value in order to reduce the global searching space.

$$\min Z = \alpha \frac{C_1 - C_1^{\min}}{C_1^{\max} - C_1^{\min}} + (1 - \alpha)\frac{C_2 - C_2^{\min}}{C_2^{\max} - C_2^{\min}} \tag{18}$$

Where $C_1^{\min}$ and $C_2^{\min}$ are the minimum values of two subobjectives, $C_1^{\max}$ and $C_2^{\max}$ are the maximum values of two sub-objectives; $\alpha$ denotes the weight of passenger-oriented subobjective, ranging from 0 to 1; $1-\alpha$ denotes the weight of operator-oriented subobjective.

## Solving procedure

The problem is structured as an MIP model, which can be solved by Lingo, a linear interactive optimizer. In Lingo, the coding is divided into the set segment, data segment, objective segment, and constraint segment. The set segment defines the sets of decision variables and data matrix, for the convenience of subsequent parameter fetching. The date segment identifies the input parameter data from the defined data sets, including the OD matrix, station spacing, modal characteristics, and other related information. The objective and constraint segments are the main body of the MIP model. Fig 2 demonstrates the solution procedure of proposed model for optimizing the multi-modal service schemes on the urban-rural bus corridor.

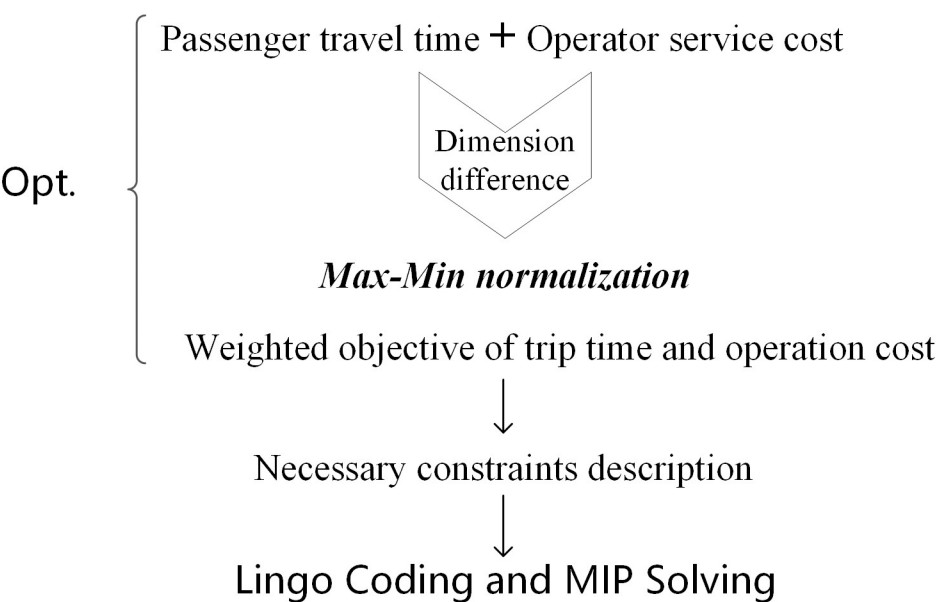

**Fig 2. The Lingo solving framework for the multi-modal optimization of urban-rural transportation.**

## Model validation and discussion

In order to validate the feasibility of the proposed optimization model, a case study was performed on the urban-rural bus corridor from Jasmine Ecological Park to West Xiangyang Bridge in Yangzhou, Jiangsu Province, China.

### Data input and parameter calibration

**(1) Route parameters.** The selected bus corridor origins from Jasmine Ecological Park in Hanjiang District, and ends at West Xiangyang Bridge in Songqiao town, connecting the urban district to suburban areas including Huaisi Town, Fangxiang Town, Gongdao Town and Songqiao Town, with a total route length of 23.77 km. All-stop buses, express buses and shuttle buses are currently operating on this corridor, under a mixed management of government sectors and transport enterprises, which lacks coordination and need joint optimization. After rejecting some dilapidated stations, the total number of active stations have been reduced to 16, as shown in Fig 3, where the station spacing distances range from 670 to 2900 m.

**(2) Passenger demand parameters.** According to our investigation, basic passenger exchange data can be obtained by conducting on-board passenger inquiries and counting station volumes along the studied bus corridor. To enhance the generality of passenger OD volume distribution, the original data is modified by factoring in the impact of surrounding land use on passenger production and attraction within a station coverage radius of 800 m. The modified passenger OD distribution is indicated in Figs 4 and 5 shows the corresponding section volume distribution along the bus corridor. The total number of passenger exchange during the peak hour is 742 pax/h, trips between stations range from 0 to 42, and the station 9 (*Changsheng Station*) owns the highest passenger generation of 120 pax/h, the station 14 (*Xiazhuang Station*) owns the highest passenger attraction of 164 pax/h.

**(3) Bus operation characteristics.** In Table 2, the operation characteristic parameters of different bus modals are indicated, and each characteristic is calibrated according to practical conditions. Taking the unit operation cost for the all-stop bus modal as an example, the unit

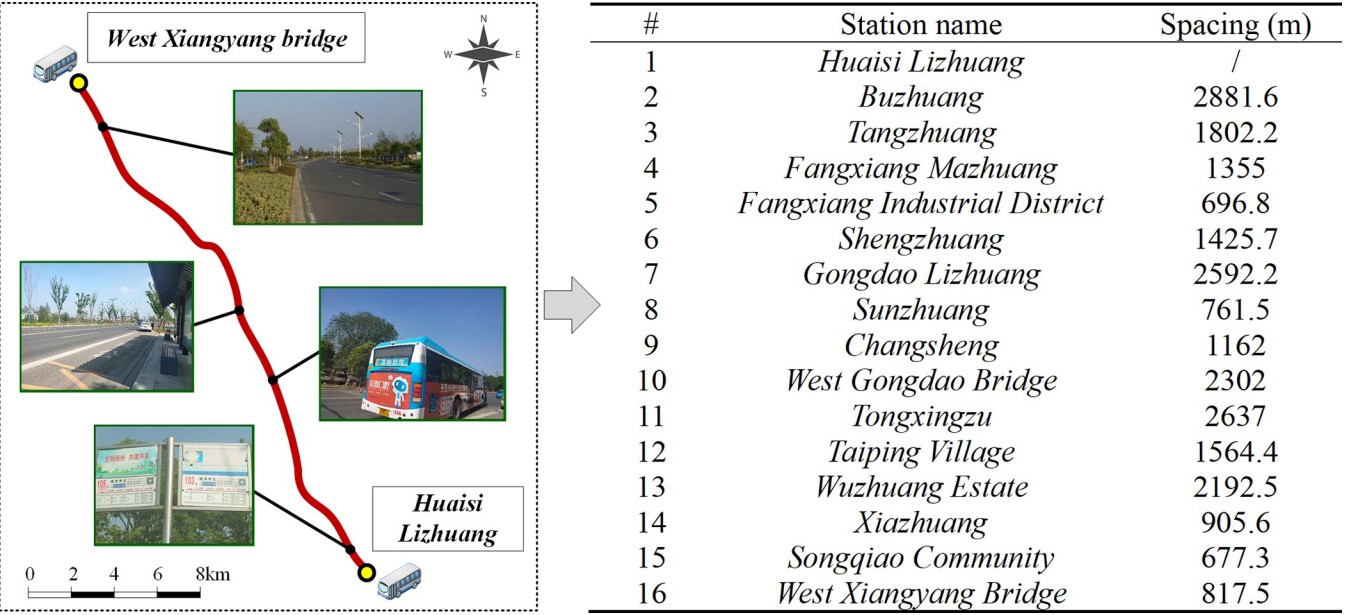

| # | Station name | Spacing (m) |
|---|---|---|
| 1 | *Huaisi Lizhuang* | / |
| 2 | *Buzhuang* | 2881.6 |
| 3 | *Tangzhuang* | 1802.2 |
| 4 | *Fangxiang Mazhuang* | 1355 |
| 5 | *Fangxiang Industrial District* | 696.8 |
| 6 | *Shengzhuang* | 1425.7 |
| 7 | *Gongdao Lizhuang* | 2592.2 |
| 8 | *Sunzhuang* | 761.5 |
| 9 | *Changsheng* | 1162 |
| 10 | *West Gongdao Bridge* | 2302 |
| 11 | *Tongxingzu* | 2637 |
| 12 | *Taiping Village* | 1564.4 |
| 13 | *Wuzhuang Estate* | 2192.5 |
| 14 | *Xiazhuang* | 905.6 |
| 15 | *Songqiao Community* | 677.3 |
| 16 | *West Xiangyang Bridge* | 817.5 |

**Fig 3. The route and active station spacing of studied urban-rural bus corridor.**

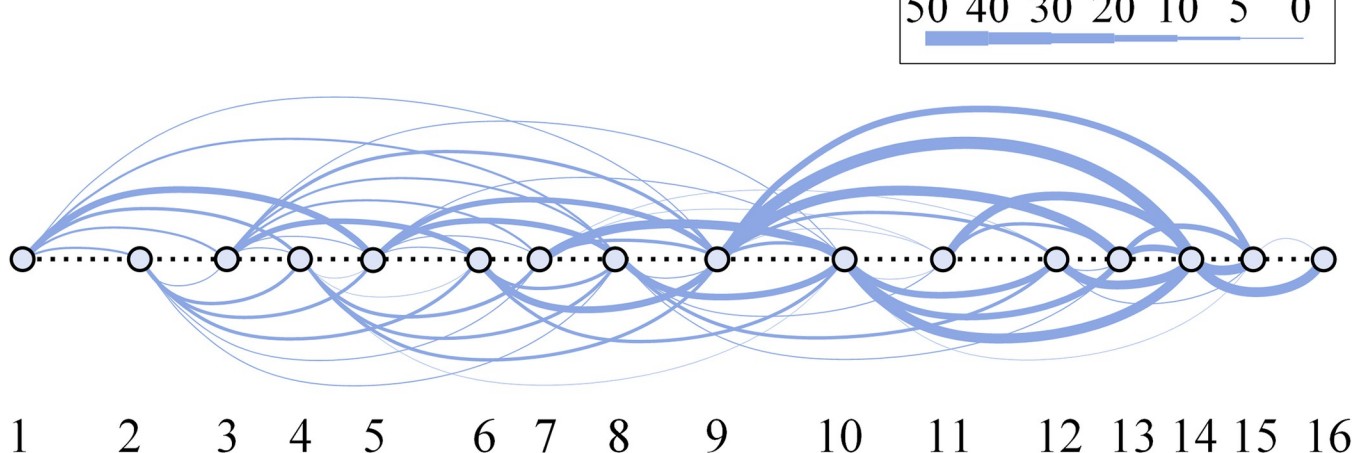

**Fig 4. The bus OD trips distribution of the corridor during the peak hour.**

time vehicle depreciation is about 41.1 Yuan/h, the electricity consumption is about 15 Yuan/h, the hourly driver wage is 50 Yuan/h, and therefore the unit operation cost for an all-stop bus on the corridor is about 106.1 Yuan/h. As for the time consumption by stopping at a station ($t_s$), the speed loss time caused by decelerating and accelerating near station $s$ is about 12 s, the door opening and closing time of bus vehicle is about 5 s, the average boarding and alighting time is about 8 s, therefore the parameter $t_s$ takes the value of 25 s.

The values of average running speed in Table 2 are calculated considering the aggregate delay time at intersections based on real observed data, as listed in Table 3. Note that the additional time consumption caused by stopping at one station (average stop delay) is 25 s ($t_s$).

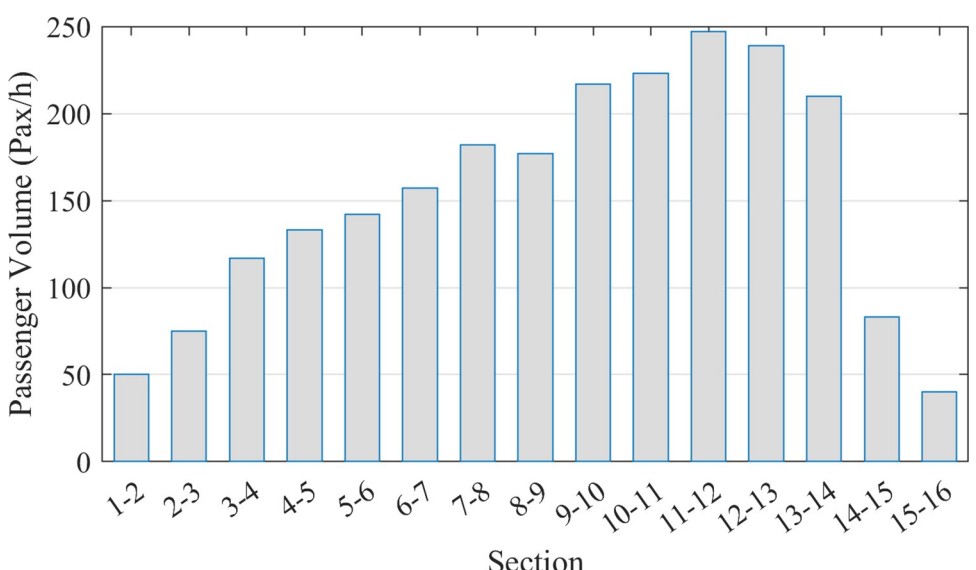

**Fig 5. The distribution of corridor section volumes during the peak hour.**

**Table 2. Input parameters of operation characteristics for three bus modals.**

| Modal | Specified vehicle capacity $b_k$ (Pax) | Average running speed $V_k$ (km/h) | Unit time operation cost $coe_k$ (Yuan/h·veh) |
|---|---|---|---|
| All-stop bus ($k = 1$) | 50 | 25 | 106.1 |
| Express bus ($k = 2$) | 35 | 35 | 101.7 |
| Shuttle bus ($k = 3$) | 20 | 30 | 79.5 |

**(4) Modal choice ratio.** Since there are three bus service modals running along the corridor, passengers will choose the bus mode they prefer based on their own expectations and income levels. E.g., the rapid bus can save the travel time due to the limited stop frequency and higher travel speed, but its ticket price is much higher, usually 2 to 4 times higher than the all-stop bus and the shuttle bus. Therefore, passengers have to judge and weigh the time-saving benefits and the ticket expenditures when choosing bus modals. Because it is hard to quantify the individual expectation choice under different trip distance, trip purpose, ticket pricing and comfortability demand, a survey has been conducted to estimate the choice probability, which was performed together with the individual attribute investigation. The aggregate sample distribution of modal choice is indicated in Table 4.

**(5) Value of time (*vot*).** Generally, *vot* varies with the income of passengers, for example, the *vot* of a farmer is usually lower than an engineer in China. Because it is hard to know the vocation of all passengers, the *vot* for urban-rural bus passengers is taken as 0.3 Yuan/(pax·min) upon a questionnaire analysis of 77 investigated passengers, represented sample distribution are listed in Table 5.

## Model outputs

Since the integrated objective assigns different weights to the costs of passenger travel time and the operator service, the optimal scheme changes accordingly. Table 6 lists the optimal results of the normalized weighting cost, the passenger travel time cost and the operator service cost under representative weight combinations, where the independent optimization of each sub-objective is equivalent to the corresponding weight value of 1. E.g., the independent optimization result of passenger travel time cost is proportional to the joint optimization result when its weighting coefficient $\alpha$ takes 1. Similarly, Table 7 lists the optimal schemes of multi-modal bus operation under different weight assignment. Meanwhile, it should be noted that

**Table 3. Running speed observation of represented bus modals.**

| Service model | Bus route | Observed running distance (km) | Observed time (min) | Intersection delay (min) | Stop delay (min) stops | Stop delay (min) time | Running time (min) | Travel speed (km/h) |
|---|---|---|---|---|---|---|---|---|
| Suttle bus | 15 | 12.1 | 30.09 | 2.81 | 7 | 2.92 | 24.36 | 29.8 |
| | 15 special | 12.1 | 31.30 | 3.33 | 6 | 2.50 | 25.47 | 28.5 |
| | 105(night) | 14.98 | 36.68 | 4.12 | 9 | 3.75 | 28.81 | 31.2 |
| Rapid bus | Youyang sightseeing | 19.22 | 41.09 | 6.15 | 5 | 2.08 | 32.85 | 35.1 |
| | Songyang express | 17.54 | 37.69 | 5.52 | 4 | 1.67 | 30.50 | 34.5 |
| All-stop bus | 103 | 17.03 | 50.28 | 5.71 | 13 | 5.42 | 39.15 | 26.1 |
| | Guoyang trunk | 11.65 | 33.87 | 3.02 | 8 | 3.33 | 27.52 | 25.4 |

Table 4. Aggregate modal choice distribution at different sections.

| Parameters | Sections only covered by all-stop and express bus | | Sections covered by three modals (Shuttle bus running section) | |
|---|---|---|---|---|
| | Number of samples | Choice ratio | Number of samples | Choice ratio |
| All-stop bus choice, $\eta_1$ | 26 | 65% | 15 | 41% |
| Express bus choice, $\eta_2$ | 14 | 35% | 5 | 13% |
| Shuttle bus choice, $\eta_3$ | / | / | 17 | 46% |
| Sum | 40 | 100% | 37 | 100% |

under the hard section volume constraint (15) for shuttle bus turn-back stations, the output turn-back stations are fixed at station 9 and 14, which are in line with the shuttle bus route from Changsheng Station to Xiazhuang Station.

## Scheme discussion

**(1) Optimal scheme recommendation.** According to the optimal results under different weight assignments, it can be concluded that the departure frequency of all-stop bus and shuttle bus basically increases with the increasing weight $\alpha$ of passenger travel time, and the departure frequency of express bus will increase dramatically when the weight $\alpha$ exceeds 0.7. According to the cost data indicated in Table 6, as compared to the independent optimization results, the variation amplitude of passenger travel time cost is within 18.3%, while the variation amplitude of operator service cost is within 96.1%, meaning that the sensitivity of bus operation cost is much higher than the passenger time cost.

Meanwhile, by introducing the ticket income as an auxiliary indicator, the ticket incomes of three bus service modals under can be obtained. According to current ticket pricing strategy, the all-stop bus and shuttle bus both charge a ticket of 2 Yuan/pax, and the express bus charges under segmented prices of 5 Yuan/pax and 10 Yuan/pax. Note that passengers using preferential tickets or IC cards are ignored here, in order to ensure the comparability among schemes. The total ticket incomes under scheme 2, scheme 5 and scheme 7 are 1710 Yuan, 1567 Yuan and 1753 Yuan, respectively. Despite scheme 7 owns the highest ticket income, its operating cost is also the highest. Therefore, scheme 2 is selected as the recommendation operation scheme, with the income composition shown in Table 8.

Therefore, the optimal scheme under the weight combination (0.3, 0.7) is chosen as the recommended scheme, where the additional variation amplitudes of two sub-objectives are 18.3% and 7.23% respectively, with the average variation rate lower than the other schemes. Under

Table 5. Represented passenger income distribution.

| Vocation | Samples | Proportion | Average income (Yuan/month) | Working hours (hours/day) | $Vot$ (Yuan/min) | Average weighted $Vot$ |
|---|---|---|---|---|---|---|
| Farmer | 15 | 19.48% | 2300 | 6 | 0.21 | 0.297 |
| Worker | 21 | 27.27% | 5100 | 9 | 0.31 | |
| Waitress | 12 | 15.58% | 4500 | 6.5 | 0.38 | |
| Teacher | 3 | 3.90% | 6800 | 8 | 0.47 | |
| Engineer | 4 | 5.19% | 6000 | 8 | 0.42 | |
| Public servant | 2 | 2.60% | 5500 | 8 | 0.38 | |
| Salesman | 7 | 9.09% | 5700 | 7.5 | 0.42 | |
| Cleaner | 3 | 3.90% | 4300 | 6 | 0.39 | |
| Security staff | 4 | 5.19% | 3800 | 12 | 0.18 | |
| Student | 6 | 7.79% | 0 | 0 | 0 | |

**Table 6. Cost comparison under different weight coefficient combinations.**

| (α,β) | Normalized weighting cost | Travel time cost (Yuan) | Operator service cost (Yuan) |
|---|---|---|---|
| (0.3,0.7) | 0.1198 | 4248.23 | 707.66 |
| (0.4,0.6) | 0.1394 | 4152 | 768.25 |
| (0.5,0.5) | 0.1403 | 4094.25 | 828.84 |
| (0.6,0.4) | 0.1262 | 3880.64 | 938.73 |
| (0.7,0.3) | 0.1296 | 3880.64 | 938.73 |
| (0.8,0.2) | 0.097 | 3626.86 | 1294.07 |
| Independent optimization | / | 3592.5 | 659.96 |

the recommended scheme, the departure frequency of all-stop, express and shuttle bus modals are 4 veh/h, 1 veh/h and 4 veh/, the index of middle stop stations for express buses are 6, 7, 8, 10 and 14, and the turn-back stations of shuttle buses are station 9 and station 14, as indicated in Fig 6.

**(2) Comparison analysis.** In the field of bus operation optimization, different methods correspond to different scenarios and purposes. For the optimization of rural-bus bus operation, some methods focus on the timetable connection optimization at transfer stations in the suburban area, some methods focus on joint optimization of ticket pricing and operation scheme of a certain service modal, and some models emphasizes the adjustment of operation route and stop station [10, 24]. Specifically, for the optimization of multi-modal bus services, the application scenarios of related studies majorly target at the busses running on city arterials or connecting transportation hubs, which require different demands in punctuality, comfort-ability and convenience as compared to our method. Meanwhile, different models have different inputs and constraints. In terms of input parameters, in addition to the passenger OD trip distribution, station spacing and basic operation parameters, different optimization model has different input parameters. E.g., in some studies, bus energy consumption calls for additional inputs of route alignment, on-board travel time is estimated according to the section traffic volume and congestion index, and the passengers waiting time needs to be calibrated upon the arrival rate and service choice probability [25, 26]. Although the proposed method is not extremely accurate of efficient, the output scheme is cost-effective and the solving procedure is time-saving.

In order to validate the effectiveness and optimality of the output scheme, a comparison between the optimal scheme and the actual scheme (shown in Table 1) has been performed. Major indicators are listed in Table 9.

The comparison results can be concluded as:

**Table 7. Operation scheme comparison under different weight coefficient combinations.**

| Scheme | (α,β) | Frequency (veh/h) | | | Stop plan of express bus |
|---|---|---|---|---|---|
| | | All-stop bus | Express bus | Shuttle bus | |
| 1 | (0,1) | 3 | 1 | 4 | 6–8,10,14 |
| 2 | (0.3,0.7) | 4 | 1 | 3 | 6–8,10,14 |
| 3 | (0.4,0.6) | 4 | 1 | 4 | 9,14 |
| 4 | (0.5,0.5) | 4 | 1 | 5 | 9,14 |
| 5 | (0.6,0.4) | 5 | 1 | 5 | 9,14 |
| 6 | (0.7,0.3) | 5 | 1 | 5 | 9,14 |
| 7 | (0.8,0.2) | 5 | 5 | 5 | 5–10 |
| 8 | (1,0) | 5 | 6 | 5 | 5–10 |

**Table 8. Ticket income analysis under scheme 2 (Unit: Yuan).**

| Station ID | On-board volume | All-stop bus | | Express bus | | Shuttle bus | |
|---|---|---|---|---|---|---|---|
| | | Volume | Ticket income | Volume | Ticket income | Volume | Ticket income |
| 1 | 50 | 46 | 92 | 4 | 20 | 0 | 0 |
| 2 | 30 | 30 | 60 | 0 | 0 | 0 | 0 |
| 3 | 50 | 50 | 100 | 0 | 0 | 0 | 0 |
| 4 | 36 | 36 | 72 | 0 | 0 | 0 | 0 |
| 5 | 45 | 45 | 90 | 0 | 0 | 0 | 0 |
| 6 | 40 | 26 | 51 | 14 | 72 | 0 | 0 |
| 7 | 52 | 33 | 67 | 19 | 94 | 0 | 0 |
| 8 | 45 | 29 | 58 | 16 | 81 | 0 | 0 |
| 9 | 120 | 49 | 98 | 0 | 0 | 71 | 142 |
| 10 | 88 | 36 | 72 | 21 | 106 | 31 | 62 |
| 11 | 40 | 16 | 33 | 0 | 0 | 24 | 47 |
| 12 | 36 | 15 | 30 | 0 | 0 | 21 | 42 |
| 13 | 35 | 14 | 29 | 0 | 0 | 21 | 41 |
| 14 | 37 | 15 | 30 | 1 | 5 | 21 | 42 |
| 15 | 38 | 38 | 76 | 0 | 0 | 0 | 0 |
| 16 | 0 | 0 | 0 | 0 | 0 | 0 | 0 |
| Sum | 742 | 479 | 957 | 75 | 377 | 188 | 376 |

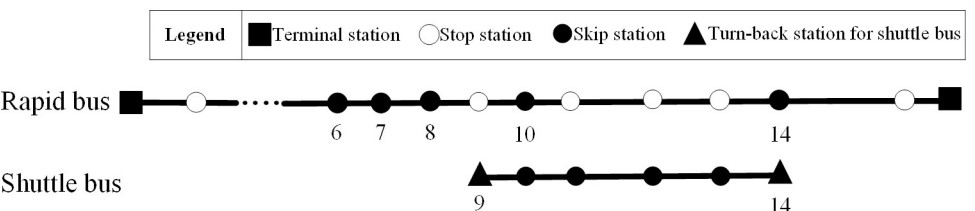

**Fig 6. Recommended stop plan for express buses and shuttle buses.**

**Table 9. Comparison between original scheme and optimized scheme during the morning peak.**

| Indicator | Service modal | All-stop | | Express | | Shuttle | |
|---|---|---|---|---|---|---|---|
| | | original | optimal | original | optimal | original | optimal |
| Scheme | Departure frequency (veh/h) | 3 | 4 (33% ↑) | 3 | 1 (67% ↓) | 5 | 4 (20% ↓) |
| | Average stops | / | / | 7 | 5 (−2) | 8 | 6 (−2) |
| Cost | Operation cost (Yuan) | 381 | 507 (33% ↑) | 124 | 38 (69% ↓) | 214 | 162 (24% ↓) |
| | Trip time (h) | 121 | 156 (39% ↑) | 38 | 21 (48% ↓) | 53 | 59 (11% ↑) |
| Section capacity | Supply (pax/h) | 150 | 200 (33% ↑) | 105 | 35 (67% ↓) | 100 | 80 (20% ↓) |
| | Average usage | 0.57 | 0.59 (4% ↑) | 0.41 | 0.54 (32% ↑) | 0.72 | 0.76 (6% ↑) |

(1) With the increase of all-stop bus departure frequency, the operation cost and capacity increase synchronously with the same increasing of 33%, while the average section capacity usage only increases by 3.5% under a small increase of passenger attraction.

(2) The decreases in the departure frequency and the average stops of express bus will cause a significant passenger volume loss, and the operation cost and trip time decrease accordingly.

(3) The decreases in the departure frequency and the average stops of shuttle bus cause a 24% decrease in the operation cost, while the trip time has increased by 11% due to the additional volume shifted from the express bus.

(4) Generally, under the optimized scheme, the total operation cost decreases from 719 Yuan to 707 Yuan, while the total trip time increases from 212 h to 246 h. From the perspective of vehicle purchasing and maintenance, the benefit of the optimal scheme outperforms the original one. Meanwhile, the average section capacity usage gets increased under the optimal scheme for all three modals.

## Conclusions

This paper highlights how all-stop, express, and shuttle bus modes along the bus corridor can be collaboratively optimized to meet the high-quality development demands of urban-rural transportation, with basic parameters of station spacing, passenger OD distribution, trip distance and further considerations on demand constraints and vehicle operation characteristics. The major contributions of this paper include:

(1) A simplified bi-objective programming model considering the traffic and trip characteristics along the urban-rural bus corridor. The simplifications are reflected in the objective formulation and parameter calibration. The objectives focus on the perceivable passenger travel time consumption and bus running cost, and the parameters are calibrated based on field observation and questionnaire investigation, in order to avoid the uncertain influences from modal service pricing, passenger preference, individual attribute, running delay and etc.

(2) An optimal scheme recommendation approach based on the sensitivity of sub-objective weights. Objective values of representative weight combinations have been discussed, together with the anticipated ticket incomes under given pricing systems, which can be provided as decision-making references for bus operators.

(3) A generalizable and executable model solving architecture. The paper presented a complete model solution method including the passenger trip characteristic analysis, the input parameter calibration, the modal service constraint formulation and the concrete code programming. Bus operators can quickly adjust the multi-modal operation scheme once the passenger trip distribution or vehicle service encounters significant fluctuations.

The proposed model is only applicable to the one-way transportation of multi-modal buses in a corridor, future research will be focused on the non-periodic operation optimization of bi-directional bus transportation and the time-varied features of passenger demand distribution. Meanwhile, it would also be interesting to study the game relationship between trip demand and service attributes (departure frequency, station spacing, vehicle type) during the cost analysis [27, 28].

## Supporting information

**S1 File. Raw dataset.** Original files of investigated data and algorithm codes.
(ZIP)

## Author Contributions

**Conceptualization:** Jun Zhang.

**Data curation:** Jun Zhang, Jingyi Qin, Jinliang Shao, Bin Lv.

**Funding acquisition:** Jun Zhang, Jiajun Shen.

**Investigation:** Jingyi Qin, Jinliang Shao.

**Methodology:** Jun Zhang.

**Resources:** Jiajun Shen, Bin Lv.

**Software:** Jinliang Shao.

**Supervision:** Jiajun Shen.

**Writing – original draft:** Jun Zhang.

**Writing – review & editing:** Jingyi Qin, Jinliang Shao.

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
