## [Decision Letter · Decision Letter 0]

9 Apr 2024

PONE-D-24-09686Collaborative Optimization of Multi-modal Transport Solutions for Urban-Rural Bus RoutesPLOS ONE

Dear Dr. Zhang,

Thank you for submitting your manuscript to PLOS ONE. After careful consideration, we feel that it has merit but does not fully meet PLOS ONE’s publication criteria as it currently stands. Therefore, we invite you to submit a revised version of the manuscript that addresses the points raised during the review process.

In the revised version of the paper please consider the reviewers' comments listed at the bottom of this email.

We look forward to receiving your revised manuscript.

Kind regards,

Camelia Delcea

Academic Editor

PLOS ONE

Journal Requirements:

"This research is partially supported by the Project of National Natural Science Foundation of China under Grant number 52302395, and partially by the MOE (Ministry of Education in China) Project of Humanities and Social Sciences under Grant number 23YJAZH122."

"Funder statement: The funders had no role in study design, data collection and analysis, decision to publish, or preparation of the manuscript.

Funding statement:

1、Project of National Natural Science Foundation of China under Grant number 52302395. Recipient: Jun Zhang；

2、MOE (Ministry of Education in China) Project of Humanities and Social Sciences under Grant number 23YJAZH122. Recipient: Jiajun Shen"

5. We note that Figure 2 in your submission contain copyrighted images. All PLOS content is published under the Creative Commons Attribution License (CC BY 4.0), which means that the manuscript, images, and Supporting Information files will be freely available online, and any third party is permitted to access, download, copy, distribute, and use these materials in any way, even commercially, with proper attribution. For more information, see our copyright guidelines: http://journals.plos.org/plosone/s/licenses-and-copyright.

Reviewers' comments:

Reviewer's Responses to Questions

**Comments to the Author**

1. Is the manuscript technically sound, and do the data support the conclusions?

Reviewer #1: Partly

Reviewer #2: Yes

2. Has the statistical analysis been performed appropriately and rigorously? 

Reviewer #1: No

Reviewer #2: Yes

3. Have the authors made all data underlying the findings in their manuscript fully available?

Reviewer #1: No

Reviewer #2: Yes

4. Is the manuscript presented in an intelligible fashion and written in standard English?

Reviewer #1: Yes

Reviewer #2: Yes

5. Review Comments to the Author

Reviewer #1: This paper proposed a collaborative optimization model for bus route. It’s very interesting. However, the authors haven’t prepared well this paper for publication in my opinion. My comments are shown in below.

1. Please list the contributions of this paper.

2. The presentation of this paper need to be improved.

3. The part of literature review need to be organized more clearly and logical. It’s not very well to summarize an article by a sentence.

4. In the section of “collaborative modelling”, authors proposed a complex optimization model with many assumptions and variables. Please illustrate a figure to show the framework of the proposed model. It’s difficult to understand the optimization method here.

5. In line 188 page 8, the variable of vot is used to calculate the time value from time cost. But the value is different in waiting, on-board or other situation in unit time. Please give explanation.

6. Public transportation network optimization should take the change of travel demands, road condition and other factors during the whole daytime into account. Here, some factors (such as road condition and so on) have a significant effect on time value calculation. Please consider

7. A bus line always has two directions, and the route of a line in different direction may be in different modal. Dose the proposed method considers bus line direction.

8. The title of this paper is “Collaborative Optimization of Multi-modal Transport Solutions for Urban-Rural Bus Routes”. However, the whole method doesn’t refer the difference between urban-rural bus and urban bus. Compared with urban, what are urban-rural bus characteristics.

9. Operator service cost is one of optimal objectives. But in practical work, the income of bus line is always an important factor in optimization. Therefore, could author add income in the model.

10. The proposed model contains a lot of parameters. Therefore, the parameter estimation is important and necessary. Please supplement the part in detail.

11. The data description is inadequate. Please describe all data collection in this paper by table and samples.

12. A comparison with other model or method should be taken in discussion, which is used to prove your model is better.

Reviewer #2: The study attempts to collaboratively optimize multi-modal buses for urban-rural bus routes. While the used bi-objective programming model is a classic model, the topic is something new. The reviewer only has a few minor comments.

1. Please clearly present the main contributions of this study.

2. For all-stop buses, express buses, and shuttle buses, their competition and cooperation relationships should be considered in the optimization.

3. Case study could be presented with more details, such as the description of the urban-rural bus corridor from Jasmine Ecological Park to West Xiangyang Bridge and a figure of such corridor.

4. The language needs more work.

6. PLOS authors have the option to publish the peer review history of their article (what does this mean?). If published, this will include your full peer review and any attached files.

Reviewer #1: No

Reviewer #2: No

---

## [Author Response · Author response to Decision Letter 0]

6 Jun 2024

Dear reviewers,

Over the past two months, we have spent great effort refining the manuscript. Meanwhile, the comments raised by Reviewers and editors have got replied and implemented. Please see the document named response letter uploaded in the system.

We are uploading (a) response document to the reviewers’ and editors’ comments (below), (b) a marked-up manuscript with indicated changes under the track changes mode in MS word, and (c) an unmarked manuscript without tracked changes.

Any further comments or requirements are welcome to us. Our deepest gratitude goes to you for your careful work. 

Thank you!

Kind regards.

---

## [Decision Letter · Decision Letter 1]

5 Aug 2024

Collaborative Optimization of Multi-modal Transport Solutions for Urban-Rural Bus Routes

PONE-D-24-09686R1

Dear Dr. Zhang,

We’re pleased to inform you that your manuscript has been judged scientifically suitable for publication and will be formally accepted for publication once it meets all outstanding technical requirements.

Kind regards,

Camelia Delcea

Academic Editor

PLOS ONE

Additional Editor Comments (optional):

Please consider amending the information in Figure S1 as suggested by the reviewer when uploading the final version of the paper. Thank you!

Reviewers' comments:

Reviewer's Responses to Questions

**Comments to the Author**

1. If the authors have adequately addressed your comments raised in a previous round of review and you feel that this manuscript is now acceptable for publication, you may indicate that here to bypass the “Comments to the Author” section, enter your conflict of interest statement in the “Confidential to Editor” section, and submit your "Accept" recommendation.

Reviewer #1: All comments have been addressed

Reviewer #2: (No Response)

2. Is the manuscript technically sound, and do the data support the conclusions?

Reviewer #1: Partly

Reviewer #2: (No Response)

3. Has the statistical analysis been performed appropriately and rigorously? 

Reviewer #1: Yes

Reviewer #2: (No Response)

4. Have the authors made all data underlying the findings in their manuscript fully available?

Reviewer #1: Yes

Reviewer #2: (No Response)

5. Is the manuscript presented in an intelligible fashion and written in standard English?

Reviewer #1: Yes

Reviewer #2: (No Response)

6. Review Comments to the Author

Reviewer #1: Thank you for your responses. The author's reply is very serious and detailed for all comments. In terms of how the article could be improved, I think it’s necessary to modified the “Constraints” of new Figure S1. It’s difficult to realize the “Variable type”. And it is also could not find in the corresponding text in “model constraints”. At the same time, the meaning of coordinate axis need to be added in Figure S1.

In general, the responds are very good. I have no other comments.

Reviewer #2: (No Response)

7. PLOS authors have the option to publish the peer review history of their article (what does this mean?). If published, this will include your full peer review and any attached files.

Reviewer #1: No

Reviewer #2: No

---

## [Editor Report · Acceptance letter]

7 Aug 2024

PONE-D-24-09686R1 

PLOS ONE

Dear Dr. Zhang, 

I'm pleased to inform you that your manuscript has been deemed suitable for publication in PLOS ONE. Congratulations! Your manuscript is now being handed over to our production team.

Kind regards, 

on behalf of

Dr. Camelia Delcea 

Academic Editor

PLOS ONE